# Strategies to Convert Cells into Hyaline Cartilage: Magic Spells for Adult Stem Cells

**DOI:** 10.3390/ijms231911169

**Published:** 2022-09-22

**Authors:** Anastasiia D. Kurenkova, Irina A. Romanova, Pavel D. Kibirskiy, Peter Timashev, Ekaterina V. Medvedeva

**Affiliations:** 1Institute for Regenerative Medicine, Sechenov First Moscow State Medical University (Sechenov University), 119991 Moscow, Russia or; 2World-Class Research Center “Digital Biodesign and Personalized Healthcare”, Sechenov First Moscow State Medical University (Sechenov University), 119991 Moscow, Russia

**Keywords:** adult stem cells, chondrogenic differentiation, tissue markers, osteoarthritis, articular cartilage, autologous chondrocytes, bone marrow, synovium, adipose tissue, dental pulp, periosteum, perichondrium, extracellular vesicles

## Abstract

Damaged hyaline cartilage gradually decreases joint function and growing pain significantly reduces the quality of a patient’s life. The clinically approved procedure of autologous chondrocyte implantation (ACI) for treating knee cartilage lesions has several limits, including the absence of healthy articular cartilage tissues for cell isolation and difficulties related to the chondrocyte expansion in vitro. Today, various ACI modifications are being developed using autologous chondrocytes from alternative sources, such as the auricles, nose and ribs. Adult stem cells from different tissues are also of great interest due to their less traumatic material extraction and their innate abilities of active proliferation and chondrogenic differentiation. According to the different adult stem cell types and their origin, various strategies have been proposed for stem cell expansion and initiation of their chondrogenic differentiation. The current review presents the diversity in developing applied techniques based on autologous adult stem cell differentiation to hyaline cartilage tissue and targeted to articular cartilage damage therapy.

## 1. Introduction

Articular cartilage has a complex structural tissue consisting of several layers, which differ by their composition in the collagen-type content, collagen fiber orientation and cell morphology and density [1] (Figure 1b). Such a complex hierarchical structure is fundamental to the unique biomechanical properties of articular cartilage [1] and, simultaneously, a challenge for regenerative medicine. However, finding a solution is essential since degenerative joint diseases are widespread. Osteoarthritis (OA) is the most common joint disease, with progressive degeneration of articular cartilage and subsequent slow joint space narrowing, pathological remodeling of all joint tissues, including bone, and chronic inflammatory processes accompanied by movement limitation [2,3,4]. Clinical remediation is urgently required, with a growing list of developing drug treatments for OA being unsuccessful in clinical trials [5].

Unfortunately, there is no better method for patients with severe knee articular cartilage damage than total joint arthroplasty. Prevalent clinically available cartilage repair surgical techniques undertaken for minor osteochondral defects (<4 cm^2^) [6,7] are microfracture and mosaicplasty, leading to fibrocartilage neo-tissue formation. Successful restoration depends on the age and size of any cartilage defects, with the data reporting that mosaicplasty is the more durable of the two applied methods [8]. Currently, ACI method and its modification, known as the matrix-induced ACI (MACI) method, are used in clinical practice [6,7]. Twelve years of investigation have confirmed that MACI might be suitable for medium-term cartilage repair [9]. However, all mentioned treatments have limited capacity to induce long-term cartilage regeneration, resulting in mechanically inferior fibrocartilage degraded with time [6,7].

The demand of reconstructing the highly organized hyaline cartilage promotes further tissue engineering research based on different adult stem cell populations, which differ significantly in their proliferative properties and chondrogenic differentiation ability [10]. As a result, the development of approaches to stimulate stem cells’ differentiation from diverse sources will vary (Figure 2).

## 2. Markers for Hyaline Cartilage Chondrocytes, Hypertrophic Chondrocytes and Fibrocartilage

Hyaline cartilage is the tissue of clinical interest. On the contrary, the hypertrophic and fibrocartilage chondrocyte phenotypes are associated with OA development [5]. During the late stage of OA disease progression, chondrocytes change their phenotype [11] to the hypertrophic and fibrocartilage chondrocytes that catalyze the destruction of articular cartilage tissues [12]. For further discussion, it is necessary to note which markers (Figure 1a) allow the distinction of hyaline cartilage chondrocytes from fibrocartilage cells, as well as from hypertrophic chondrocytes, which tend to be replaced by bone [13].

The fate of cartilage existence depends upon the induction of some genes and suppression of others. Chondroprogenitors differentiate into mature hyaline cartilage chondrocytes producing cartilage-specific matrix proteins, such as type II collagen (COL2) and aggrecan (ACAN). Other hyaline cartilage ECM proteins are collagen types IX (COL9) and XI (COL11), but they are less abundant than COL2 and ACAN [13,14].

Chondrocytes undergo hypertrophic differentiation characterized by the secretion of specific ECM molecules [15], such as collagen type X (COL10), the most well-known hypertrophy marker [16]. In vitro analyses demonstrate direct regulation of COL10 gene (*Col10a1*) expression by myocyte-enhancer factor 2C (MEF2C) [17]. Genetic deletion of *Mef2C* impairs hypertrophy, cartilage angiogenesis and cartilage following endochondral ossification, preventing longitudinal bone growth in mice [17]. MEF2C seems to regulate another hypertrophic marker belonging to the RUNX family of transcription factors, RUNX2 (also known as CBFA1). RUNX2 and another family member, RUNX3, are necessary for chondrocyte hypertrophy, matrix mineralization and osteoblast differentiation [15,18,19].

On the contrary, RUNX1 suppresses the hypertrophic differentiation of chondrocytes [20], enhancing hyaline cartilage matrix production in cooperation with SOX-family molecules [20,21], which regulate chondrogenesis [21] and are conserved across vertebrate taxa [22,23]. Chondrocytes and chondroprogenitors express SOX9, a major chondrogenic transcription factor, and its cofactors SOX5 and SOX6, which stimulate ECM gene expression, including genes of COL2 and ACAN [14,21]. Another highly conserved mammalian molecule, a transmembrane chondroitin sulfate proteoglycan, named secondary ossification center associated regulator of chondrocyte maturation (Snorc), was described as a mature chondrocyte phenotype marker and restricted to hyaline cartilage structures [24]. *Snorc* gene expression is mediated through the SOX9 enhancer in vitro [25] and in vivo [14]. One more protein regulated directly by SOX9 is a ubiquitin ligase, WW domain-containing protein 2 (WWP2) [26]. The sequence of the *Wwp2* gene includes the gene of microRNA-140 (miR-140). Both WWP2 and miR-140 are abundantly and specifically expressed in cartilage tissues [26,27] and maintain cartilage homeostasis via RUNX2 poly-ubiquitination and degradation [28].

The fibrocartilage chondrocyte phenotype is mainly found in joints in the late stages of OA. It is characterized by increased expression of collagen types I (COL1) and III (COL3), as well as α-smooth muscle actin (αSMA), fibroblast-specific protein 1 (FSP1 or S100A4), tissue inhibitor of metalloproteinase-1 (TIMP1) and cell migration inducing hyaluronidase-1 (CEMIP) (Figure 2) [29].

In most cases, it is enough to demonstrate the absence of the COL1 protein to exclude the presence of fibrocartilage. A classical set of molecules confirming chondrocytes’ hyaline cartilage state are SOX9, COL2 and ACAN. COL10, MEF2C and RUNX2 presence are usually analyzed to reveal hypertrophy. Other markers are less commonly reported in publications.

## 3. Autologous Chondrocytes from Different Sources

ACI and MACI procedures are approved for clinical application. However, these therapies have significant drawbacks. In the late stages of OA, extraction of the autologous chondrocytes is no longer an available option. Another widespread problem is that the obtained chondrocytes are inclined to lose their chondrogenic features (de-differentiate) in monolayers [30].

### 3.1. Chondrons, the Functional Units of Cartilage Tissue

To avoid chondrocyte de-differentiation in vitro, it was suggested to isolate chondrons composed of chondrocytes surrounded by a pericellular matrix. A milieu of chondrocytes play an essential role in the maintenance of their chondrogenic state and utilization of chondrons could retard chondrocyte de-differentiation processes in vitro [31]. A recent study demonstrated that cultured chondrons expressed higher levels of proteoglycan and COL2 than chondrocyte culture [32]. Surprisingly, a mixture of chondrons and chondrocytes significantly elevated the production of ECM in culture even more than the chondrons alone. Implantation of alginate spheres, including chondrons and chondrocytes, accelerated the regeneration of injured knee cartilage in rabbits [32].

The extraction of chondrons from cartilage is not more difficult than chondrocyte isolation [32,33]. However, X-ray micro-computed tomography (μCT) methods showed that the morphology of chondrons is affected by OA progression [34]. The limitations of using autologous chondrons to repair a cartilage defect are the same as for autologous chondrocytes.

### 3.2. Chondroprogenitors from Superficial Zone of Articular Cartilage

One way to avoid chondrocyte de-differentiation during expansion is to use chondrogenic progenitors, which are cells with an incomplete differentiation cycle but a higher ability to proliferate. A population of endogenous, slowly dividing progenitor cells was discovered in the superficial zone of adult articular cartilage [35,36]. Recently, it was confirmed that these articular cartilage progenitor cells (ACPCs) in the superficial zone were responsible for the turnover of articular cartilage tissue in the postnatal period in mice [36,37].

Since ACPC cultures have not been scrupulously investigated yet, there are no unique cell markers for the selective isolation protocol, especially for human ACPCs. Different unsuccessful efforts have been made [38,39] to set apart the ACPCs and mature chondrocytes; however, the method of ACPC isolation, which is predominant in publications but has low specificity, relies on adhesion to a fibronectin coating (owing to the high cell expression of the fibronectin receptors integrin-α5 and -β1) [35].

To determine the optimal chondrogenic conditions, bovine ACPCs were cultured as pellets (a high-cell-density scaffold-free culture compressed by centrifugation) for 21 days in the presence of known chondrogenic factors, such as transforming growth factor-β (TGFβ) 1 to 3, bone morphogenetic proteins (BMP) 2 and 9, synthetic glucocorticoid dexamethasone, C-natriuretic peptide, dibutyryl-cAMP, concanavalin-A, ethanol and chelerythrine chloride [40]. Histological analysis, immunohistochemical (IHC) labeling, polymerase chain reaction (PCR) and atomic force microscopy measurements revealed BMP9 as a promising chondrogenic factor for ACPC differentiation in vitro. However, the concentration of BMP9 in the culture medium should not exceed 100 ng/mL to avoid the production of the hypertrophic marker COL10 [40].

Replacing the routinely used growth supplement fetal bovine serum (FBS) with human platelet lysate in the culturing protocol led to faster doublings and increased expression of ACAN and COL2, as well as COL10 and COL1, in ACPC monolayer cultures [41]. Compared to monolayers, the technique of ACPC expansion on commercial macroporous gelatin-coated microcarrier beads in the presence of TGFβ1 resulted in a significant enhancement in ACPC proliferation, maintaining the ability to produce an ECM abundant with COL2 and glycosaminoglycans (GAGs) [42].

The articular cartilage chondrocytes are well-specialized cells that are sensitive to mechanical stimuli, which are a complex combination of compression, stretching, shear stress and hydrostatic pressure. Mechanical loading increased SOX9 IHC labeling in the superficial zone of bovine cartilage cultured as cylindrical osteochondral plugs [43]. The application of bioreactor-generated joint-like movements demonstrated the mechanosensitivity of ACPC cultures [44]. Intermittent hydrostatic pressure (IHP) applied to ACPCs seeded within alginate beads for four weeks stimulated more significant production of cartilage ECM components and the expression of chondrocyte-related genes compared to cultures of adipose-derived stem cells (ADSCs) or even chondrocytes [45]. Mechanical stimulation is assumed to be more relevant to chondroprogenitors than chondrocyte cultures.

During in vivo studies, a construct consisting of goat ACPCs seeded on a COL1/COL3 membrane (Chondro-Gide^®^) was transplanted into a knee defect. Analysis showed acceptable integration with the surrounding cartilage and COL2-positive staining of neo-tissue. However, no difference was found in the final results between ACPC- and chondrocyte-treated defects [46]. During a clinical trial, human CD146-positive cells collected from the upper zone of articular cartilage were differentiated to a chondrogenic lineage in a culture medium containing TGFβ3 and BMP4 [47] and surgically implanted on a COL1/COL3 membrane into knee cartilage defects. Subsequent magnetic resonance imaging (MRI) and arthroscopy showed complete attachment of the construct and filling of the defect, with histology and IHC estimation of neo-tissue confirming a good repair [47]. However, the significant regeneration was believed to have resulted from the young age of the patients treated (the average age was 25 years).

### 3.3. Non-Articular Autologous Chondrocytes

To exclude the donor-site morbidity of a joint during autologous articular chondrocyte (AC) isolation, it is suggested that non-articular chondrocytes be used for joint defect treatment. The efficacy of replacing hyaline cartilage cells with non-articular chondrocytes remains debatable.

#### 3.3.1. Nasal Chondrocytes

Both articular and nasal tissues are permanent hyaline cartilage throughout life but have different origins during embryonic development. Nevertheless, in adults, nasal cartilage tissue has a similar composition and structure to joint cartilage, but the calcified zone is not present in nasal cartilage [48]. The monolayer of adult nasal chondrocytes (NCs) has three-times faster proliferative activity and promising chondrogenic capacities [49]. In an animal defect model, NCs within engineered grafts could generate hyaline cartilage markers even more than ACs [50].

Engineered autologous human cartilage grafts obtained from a nasal septum biopsy specimen were implanted into femoral full-thickness cartilage defects in a first-in-human trial, including ten patients [51]. IHC characterization of the non-used portions of the engineered cartilage constructs indicated positivity for hyaline cartilage markers, slight positivity for COL1 and negativity for COL10. Twenty-four months after the surgery, histological analysis of the biopsy at the engineered graft implantation site indicated heterogeneity in the repair tissue without the typical hyaline cartilage organization. However, the high patient self-assessment scores and MRI analyses of the defect filling allowed the authors to claim a satisfactory clinical outcome [51].

A pre-clinical study from 2021 showed that nasal-chondrocyte-based tissue-engineered cartilage (NTEC) implanted into osteochondral cartilage defects of sheep prevented the elevation of synovial fluid volumes and decreased concentrations of pro-inflammatory cytokines induced by knee OA. Autologous NTEC graft implantation into a human was evaluated in two patients with advanced OA. Fourteen months after the surgery, the patients confirmed improvements in pain, knee-joint function and quality of life. Accordingly, the authors declared that future controlled trials on other joints would be initiated [52].

#### 3.3.2. Costal Chondrocytes

Costal hyaline cartilage is another chondrocyte source for ACI and MACI therapies, allowing for potential complications following additional joint surgery interventions to be avoided. Moreover, autologous costal cartilage is already widely used as a source of grafts in plastic surgery, demonstrating the procedure’s safety.

During culturing without growth factors, costal-derived chondrocytes (CCs) proliferate 9-fold faster than ACs. However, both the ACs and CCs gradually lose their phenotype following the first four passages [53]. The addition of fibroblast growth factor 2 (FGF2) in combination with commercial mesenchymal stem cell growth medium^TM^ (MSCGM) induced stromal cell features with a significant content of COL2 and ACAN, with trace amounts of COL1. At the same time, FGF2 and DMEM drove CCs to transition to fibroblast-like cells [54]. In a rabbit osteochondral defect model, CCs expanded in the FGF2-containing MSCGM, resulting in good defect healing [54].

A clinical test of autologous CC pellets with fibrin glue on seven patients with full-thickness articular cartilage lesions included CCs expanded in MSCGM with FGF2 before implantation [55,56]. The results showed significant improvements in all clinical scores during the 5-year follow-up period confirmed by MRI scans. Nevertheless, hypertrophy in implants or incomplete defect filling was observed in two patients [55]. According to recently published data, co-culturing CCs with synovial membrane-derived mesenchymal stem cells (SM-MSCs) could prevent hypertrophy and maintain the chondrogenic phenotype of CCs in cartilage repair [57].

#### 3.3.3. Growth Plate Chondrocytes

The epiphyseal/growth plate hyaline cartilage cells are responsible for the elongation of long bones. The age of growth plate closure for humans is 17–25 years old [58]. Cell harvesting could damage the immature growth plate and disrupt bone growth [59] and it severely limits the use of growth plate chondrocytes for autologous transplantation. However, porcine chondrocytes from proliferating (middle) zones of growth plate cartilage have been isolated and characterized in vitro [60] as a more viable, proliferative and preserved chondrogenic phenotype that lasts longer during passages compared to ACs [60].

It has been previously assumed that an avascular hyaline cartilage nature provides “immune privilege”, inferring the potential of using the growth plate as a donor source in the case of allogeneic (cadavers or amputated material) or xenogeneic transplantation. This notion was disproved by researchers [61] when the rabbit model showed that, in contrast to allogeneic transplantation, xenogeneic implants in joint cartilage provoked a marked innate and adaptive immune response [61].

For obvious reasons, at present, studies about the application of growth plate chondrocytes to articular cartilage defect treatment are scarce.

#### 3.3.4. Chondrocytes from the Auricle of an Ear

In contrast to joint cartilage, auricular cartilage is covered by a perichondrium, contains elastic fibers and COL1 and does not show a defined zonality in its structure. Auricular chondrocyte (AUC) yields per tissue volume are high and sufficient for cell therapy realization. De-differentiation has been reported to proceed more rapidly in AUCs than in ACs, which was closely related to their proliferative activity [62,63].

Wong et al. (2018) suggested converting the de-differentiated monolayer AUCs to hyaline cartilage using ECM [64]. The data demonstrated that the COL-coated culture dish promoted hyaline cartilage markers and suppressed auricular elastic cartilage markers (such as COL1 and elastin) in rabbit AUCs compared to cultivation on plastic. Three months after implantation into an osteochondral defect, AUCs cultured in COL2 scaffolds exhibited successful integration with the articular cartilage, abundant proteoglycan syntheses and intense staining for COL2 with traces of elastin staining [64]. It was noted that the AUCs without the COL2 scaffold did not result in compatible healing outcomes.

Wang et al. (2018) suggested an original cell-free approach combining AUC sheets and the microfracture technique [65]. Rabbit AUC sheets were gently decellularized, retaining the native architecture and ECM composition. Decellularized AUC sheets tested in vitro showed stimulation of potential migrating BM-MSCs, increasing SOX9 production and decreasing COL10 production detected by RT-PCR and Western blots [65]. Agreeable results for the application of decellularized AUC sheets in combination with microfracture to osteochondral knee defect regeneration in rabbits were confirmed by macroscopic observation, μCT imaging and histological analysis 12 weeks after the surgery [65].

Goat AUC sheets differentiated in a chondrogenic medium supplemented with TGFβ1, insulin-like growth factor 1 (IGF1) and dexamethasone indicated a gradual increase in hyaline-cartilage-specific ECM analyzed by histological and IHC staining [63]. However, after subcutaneous implantation of AUC sheets in nude mice, a threat of AUC sheet tissue ossification was demonstrated [63].

The clinically available growth factors FGF2 and IGF1 and the peptide hormone insulin were proposed as optimal for AUC expansion in monolayer cultures [66]. Their effect on the differentiation of three-dimensional (3D) cultures of AUCs was investigated [67]. Histological, biochemical and biomechanical analyses showed that IGF1 stimulated the chondrogenesis of AUC 3D constructs in vitro better than the other molecules tested. However, the chondrogenic effect was not confirmed after subcutaneous transplantation into nude mice [67].

Easy access to the auricle cartilage makes this cell source too attractive for regenerative medicine to give up trying to develop a strategy involving AUCs in cartilage defect regeneration.

## 4. Bone-Marrow-Derived Mesenchymal Stem Cells

Beyond chondrocytes, BM-MSCs are a promising cell source for their proliferative capacity and repair potentials for both cartilage and bone tissues. Historically, MSCs were first explicitly described in the bone marrow [68,69]. In 2006, the first criteria for defining MSCs were proposed by the International Society for Cellular Therapy (ISCT) and included (a) proliferation ability and colony formation, (b) trilineage differentiation capacity and (c) the presence of the cell surface markers CD271, CD44, CD105, CD73 and CD90 and the complete absence of CD45, CD34, CD11b, CD19 and human leukocyte antigen-DR isotype (HLA-DR) membrane proteins [70]. Since then, due to new knowledge about stem cells, the criteria have been updated and become wider and less rigorous. According to the new ISCT position statement, MSCs from different source tissues could have different surface markers [71,72] and some negative markers should also be reconsidered, since some populations are characterized, for example, by the expression of CD34 or HLA-DR [71]. In addition, for the use of the term “stem” cells, it is important to confirm the self-renewal of cells in vivo or in vitro [71]. Finally, the presence of immunomodulatory properties and the characterization of trophic factors secreted by MSCs and some miRNAs have been recommended to become mandatory criteria [71,73,74].

Despite the extensive study of bone cells, there is still no unique technique to isolate a homogeneous population of BM-MSCs [75]. Cultivation of BM-MSCs of several passages increases the homogeneity of a population, but not sufficiently. In addition to plastic adhesion, negative selection of BM-MSCs was proposed; however, this increased the cost of the procedure and the purity of the stem cell population was still low [76]. Currently, selection using positive markers is preferable. Nonetheless, an isolated CD271-positive population of BM-MSCs [77] included several BM-MSC subpopulations according to single-cell RNA sequencing (scRNA-seq) analysis [78]. Thus, to date, BM-MSC heterogeneity has made stem-cell-based therapy outcomes challenging to predict.

### 4.1. Fundamental Growth Factors for BM-MSC Chondrogenic Differentiation

Currently, it is well established that the coordination of vertebrate limb development during embryogenesis depends on the interaction of the WNT and FGF families of signaling proteins [79]. The presence of FGF signals (FGF2 or FGF8) specifically countered the irreversible blocking of SOX9 expression caused by WNT activity (WNT3A) in the limb cells of embryos [79,80]. Human BM-MSCs (hBM-MSCs) cultured with a combination of Wnt3a and FGF2 supported extensive cell expansion over multiple passages and improved subsequent chondrogenic differentiation, increasing GAGs and COL2 content [81]. During late-stage chondrogenesis, active WNT signaling can promote osteogenesis [82]. Addition of the WNT inhibitor IWP2 for the last weeks of culturing reduced the content of hypertrophic and osteogenic markers in differentiating BM-MSCs [81].

Another fundamental factor involved in skeletal development is a member of the TGFβ superfamily, a pro-chondrogenic factor in embryogenesis [83] known as growth and differentiation factor 5 (GDF5, also CDMP-1 or BMP14). It is recognized that all mature joint components originate from *Gdf5*-expressing joint progenitors condensed in the interzone during early joint development [84,85]. GDF5 overexpression in BM-MSCs cultured for two weeks demonstrated significant chondrogenic effects, evidenced by SOX9 and COL2 expression, without changes for COL1 and COL10 [86]. A 24-week in vivo study on a rabbit defect model showed that an implanted GDF5-conjugated scaffold supported the chondrogenic differentiation of BM-MSCs and stimulated the complete filling of the defect with a tissue comprising proteoglycans and COL2 [86]. Intra-articularly injected GDF5-transfected autologous BM-MSCs demonstrated a similar effect according to the high histological score assessment [87]. In contrast, another publication reported that GDF5 induced the tenogenic differentiation of BM-MSCs and a significant down-regulation of non-tenogenic marker genes, such as *SOX9* [88].

The pellets of BM-MSCs demonstrated high kinetic growth for at least ten passages during culturing with members of the TGFβ superfamily, namely, TGFβ3 and BMP2 and synthetic glucocorticoid dexamethasone [89]. Comparative data based on studies of hBM-MSCs cultured with different glucocorticoids demonstrated that the glucocorticoid receptor interaction enhanced TGFβ3-induced phosphorylation of Smad2 and Smad3 [90], leading to subsequent mTORC1/AKT pathway activation and marked potentiation of transcriptional gene activity for *SOX9*, *COL2a1* and *COL10a1* [91]. Under the same conditions, dexamethasone demonstrated a substantially weaker chondrogenic effect in combination with TGFβ3 than glucocorticoid fluocinolone acetonide [91].

### 4.2. External Stimuli Supporting Chondrogenic Differentiation of BM-MSCs

He et al. [92] cultured BM-MSCs for 12 weeks on a polyglycolic acid/polylactic acid (PGA/PLA) scaffold in a serum-free chondrogenic medium containing TGFβ1, IGF1 and dexamethasone, providing relatively mature hyaline cartilage-like tissue with superior mechanical properties but a high content of COL1 and COL10. However, BM-MSCs cultured on the PGA/PLA scaffold that were implanted into an osteochondral defect in a swine model exhibited a higher cell density than the surrounding native cartilage, with mature lacuna structures and satisfactory integration into both cartilage and subchondral bone tissues [92]. The authors believe an in vitro step is needed to exclude the influence of the OA environment of the joint during the construct maturation process. This hypothesis is supported by a study where equine BM-MSCs overexpressing anti-inflammatory IL-10 did not prevent degradation of the ECM of cartilage explants in an in vitro OA model [93].

Within the articular cartilage microenvironment, endogenous growth factors, mechanical stimulation and a hypoxic environment are favorable factors that support chondrogenic cell properties. Chondrogenic differentiation of BM-MSCs induced by a combination of dexamethasone and TGFβ3 under hypoxia leads to increased expression of chondrogenic markers compared to normoxia. However, transplantation of hypoxic BM-MSC cultures into ovine articular cartilage defects produced no benefits compared to normoxia [94]. BM-MSC micro-pellets cultured under hypoxia in a chondrogenic medium were stained for GAG but showed elevated COL10 and RUNX2 levels compared to AC micro-pellets [95]. Generally, hypoxic conditions have pro-chondrogenic and anti-inflammatory impacts on stem cell cultures, while negative effects on cell viability are not observed [96].

Physical stimuli, such as matrix stiffness and hydrostatic pressure, are fundamental regulators of MSC fate. Cyclic pressure affects the composition of the ECM not only in cultured chondrocytes but also in cultured BM-MSCs, apparently through similar calcium-dependent mechanisms [97] and chondrogenic TGFβ3 stimulation of BM-MSC cultures can be modulated by the timing of the applied mechanical stimulation [98]. Applying hydrostatic pressure followed by shear stress without growth factor supplementation increases the expression of *Sox9*, *Acan* and *Col2* in cultured hBM-MSCs, compared with TGFβ1 supplementation without mechanical stimuli [99].

### 4.3. BM-MSC Transplantation Clinical Trials for Cartilage Defect Treatment

In clinical trials, intra-articular injection of autologous BM-MSCs in patients with end-stage OA was safe and showed statistically significant improvements in pain scores and quality of life [100]. However, three months after the BM-MSC injection, the cartilage catabolic biomarkers largely remained unchanged in the synovial fluid and no improvements in cartilage morphology were observed, based on MRI, one year later [100].

A study comparing the long-term clinical outcomes of ACI versus BM-MSC application was undertaken on seventy-two patients [101], where implantation of pre-expanded BM-MSCs in α-MEM supplemented with autologous serum into knee chondral defects resulted in equivalent clinical outcomes to first-generation ACI at up to 10 years [101]. Additionally, under arthroscopy, no significant differences were found between the efficacy of the transplantation of autologous BM-MSCs combined with microfracture and microfracture alone [102].

## 5. Synovial Membrane-Derived MSCs

It is currently known that heterogeneous MSC subpopulations localize in different joint regions, including the synovial membrane producing synovial fluid for lubrication of articular cartilage and its nourishment through diffusion. De Bari et al. (2001) were the first to demonstrate, on postmortem samples, that MSC-like cells isolated from human knee joint synovium by enzymatic digestion were capable of three-lineage differentiation in vitro [103,104].

Human synovial cell research, in most cases, involves synovial cells obtained during surgeries from synovium inflamed as a consequence of trauma or destructive joint disease, such as OA and rheumatoid arthritis (RA). It was identified that SM-MSCs of the OA joint maintained the ability for chondrogenic differentiation in vitro, as was the case for SM-MSCs from a healthy knee [105]. Kohno et al. (2017) found no significant differences between SM-MSC pellets from RA and OA groups [106]; after chondrogenic induction, both became transparent, intensively stained for Safranin O, showing positive immunostaining for COL2 and had comparable mRNA levels of *ACAN* and *COL2*.

### 5.1. Factors for Chondrogenic Differentiation in SM-MSCs

Pellets of SM-MSCs cultured in DMEM supplemented with TGFβ3, BMP2 and synthetic glucocorticoid dexamethasone demonstrated high kinetic growth for at least ten passages compared to the pellets from the BM and periosteum [10] and they revealed a synergistic dexamethasone-dependent (within 1–10 nM) chondrogenic effect estimated by tissue volume, proteoglycan content and COL2 expression [89]. However, a dexamethasone concentration over 100 nM promoted a minor level of adipocyte formation in SM-MSC cultures [89].

Studies indicated that BMP2 alone could promote chondrogenic differentiation along with hypotrophy and endochondral ossification [107,108]. Constitutive Smad7 expression inhibited BMP2-induced chondrogenesis in stem cells [109] and could stimulate the terminal maturation of chondrocytes [110]. In a BMP2-induced chondrogenic differentiation experiment, inhibition of Smad7 expression with short interfering RNA (siRNA) led to significantly increased expression of chondrogenic marker genes and lower expression of COL10 and matrix metallopeptidase 13 (MMP13) in human SM-MSC cultures [108].

Among others, BMP7 supplementation to SM-MSC cultures demonstrated chondrogenic differentiation of cells, as assessed by the increase in COL2, SOX9 and ACAN expression within two weeks [111]. After in vitro tests, BMP7-loaded fibrous biodegradable scaffolds with SM-MSCs were implanted into rabbit osteochondral defects. Complete healing with neocartilage tissue abundant with proteoglycans and COL2 was observed six weeks after treatment [111].

A recent study showed that tetrahedral frame nucleic acids (tFNAs), a novel DNA nanophase material, promoted the chondrogenic differentiation of SM-MSCs through Smad2/3 phosphorylation in vitro [112], which affected SM-MSC proliferation by activating the Wnt/β-catenin pathway and enhanced the migration activity of SM-MSCs in vitro [112]. Intra-articular injection of tFNAs in rabbits promoted more rapid cartilage defect regeneration with less fibrous tissue than in the control group after 12 weeks. Such outcomes suggest that tissue restoration was provided by the impact of tFNAs on the number of BM-MSCs that migrated to the defect area [112].

### 5.2. Different Cell Subpopulations Labeled in Synovium

Synovium could be histologically classified into the lining layer, sub-lining layer and perivascular region consisting of several types of cells [113,114]. An SM-MSC population that was found was not homogeneous and MSC-like cells were detected in all three regions. The obtained hSM-MSCs demonstrated different patterns of proliferation and differentiation in vitro, depending on their location in the synovium [114].

Recently, specific markers to separate hSM-MSCs according to their location in the synovium were identified [114]. After immunostaining human OA synovium sections, the markers selected were CD55^+^CD271^−^ for synovial cells in the lining layer, CD55^−^CD271^−^ in the sub-lining layer and CD55^−^CD271^+^CD90^+^ in the perivascular region [114]. The cells from the different synovial regions were cultured in DMEM supplemented with TGFβ3 and BMP2 and showed different proliferation and chondrogenic differentiation patterns in vitro [114]. The perivascular synovium cells showed large-sized pellets and higher expression levels of *SOX9*, *ACAN* and, unfortunately, *COL10* mRNA at the end of chondrogenic cultivation [114].

Fibroblast activation protein-α (FAPα) was recently confirmed as a biomarker of tissue inflammation in the lining and sub-lining layers of the synovial membrane and to be associated with a pathogenic synovial cell phenotype [115,116]. The FAPα-expressing protein in the combination of podoplanin (PDPN) and CD90 (also known as thymus cell antigen 1, THY1) enabled synovial cells in the sub-lining layer (FAPα^+^PDPN^+^THY1^+^) to be distinguished from the synovial cells in the lining layer (FAPα^+^PDPN^+^THY1^−^) in human RA tissue [116]. Injecting PDPN^+^FAPα^+^THY1^−^ or PDPN^+^FAPα^+^THY1^+^ cells into the inflamed ankle joint of mice is a different method for investigating arthritis degeneration processes. Injection of PDPN^+^FAPα^+^THY1^+^ cells resulted in more severe joint swelling with higher immune cell infiltration, with little effect on joint tissue destruction. In contrast, the injection of PDPN^+^FAPα^+^THY1^−^ cells selectively mediated bone and cartilage damage, with little effect on inflammation [116].

### 5.3. Clinical Trials of Autologous SM-MSC Transplantation into Cartilage Defects

Most in vivo research reported successful cartilage repair following SM-MSC transplantation, despite the diversity of laboratory animals and cell harvesting techniques used, strategies of cell delivery and methods for outcome estimation [117]. A 2015 clinical trial of autologous SM-MSCs was undertaken [118] using SM-MSCs expanded in α-MEM containing 10% autologous human serum for 14 days. The solution with expanded SM-MSCs was syringed into an osteochondral defect under arthroscopic control. The Lysholm score increased after treatment in each patient. Fibrous tissue was observed at the surface in three of four biopsy specimens analyzed. The sequential MRI showed that at six months and six years later, in contrast to the bone defect part, the cartilage defect was incompletely filled with tissue [118].

In 2018, implantation of a scaffold-free tissue-engineered construct generated from autologous SM-MSCs was performed on five patients in Japan [119]. SM-MSCs were enzymatically extracted from arthroscopically taken synovium and cultured for 2–3 weeks in monolayers with ascorbic acid 2-phosphate (Asc-2P), which significantly increased the self-supporting mechanical properties of the cartilage-like construct without containing artificial scaffolding [119,120]. The tissue-engineered construct was highly adherent to the normal cartilage; therefore, implantation into the defect without using sutures or fixation glue was performed. Forty-eight weeks post-surgery, according to MRI assessments and biopsy specimen histology, defect repair with abundant proteoglycan tissue and without hypertrophy occurred in all cases. The patients reported significant clinical improvement at the final 24-month follow-up [119].

## 6. Adipose-Derived Stem Cells

The essential features of adipose tissue as a cell source are the excellent stem cell yield from fat and, of course, the minimal risk for patient health while obtaining a relatively large amount of tissue [121]. Adipose-derived stem cells (ADSCs) express a profile of cell-surface molecules, similar but not identical to that of BM-MSCs [122], and ADSCs may be differentiated into mesenchymal lineages [123]. A study on rabbits demonstrated that the expansion properties and chondrogenic potential reduced with age for BM-MSCs, but not for ADSCs [124].

### 6.1. Clinical Trials of ADSC Intra-Articular Injections for Cartilage Damage Treatment

The attractive availability of ADSCs has initiated many clinical studies of the regenerative possibilities of ADSCs injected into joints to restore articular cartilage tissue lost due to OA. A meta-analysis in 2021 [125] demonstrated a statistically significant improvement in WOMAC (Western Ontario and McMaster Universities Osteoarthritis) scores after intra-articular injections of ADSCs in patients with knee OA [126], providing improved joint function and pain reduction between two months and two years after treatment [125]. However, the therapeutic effect duration is still unknown due to a lack of long-term studies. In most studies, intra-articular ADSC injections were performed on unspecified joint spaces. Only six studies included a control group and only two conducted a second-look arthroscopic assessment and subsequent histological analysis [125]. Overall, this confidently demonstrates the safety of ADSC therapy, rather than its effectiveness, in OA treatment.

A Phase IIa proof-of-concept randomized, single-blind clinical trial showed that after six months, intra-articular injection of autologous ADSCs in combination with microfracture and hyaluronic acid provided clinical improvement for patients with knee cartilage defects compared to microfractures alone [127]. The study outcomes were supported by decreased WOMAC total indexes and radiological, arthroscopic and histological evaluation during the second-look arthroscopy [127]. Complete recovery of hyaline cartilage was not observed. However, the results are sufficiently promising to encourage larger scales of randomized clinical trials.

### 6.2. Practices to Improve the Chondrogenic Potential of ADSCs

To improve the chondrogenic potential in vitro, ADSCs encapsulated with fibrin and thrombin in atelocollagen (COL1 fibers without antigenic active telopeptides) gel were cultured in chondrogenic differentiation medium supplemented with dexamethasone, insulin–transferrin–selenium (ITS) and TGFβ3 over 21 days [128]. Histological analysis of the atelocollagen scaffolds seeded with ADSCs revealed lacunae-like structures and glycosaminoglycan accumulation. Cartilage-matrix-related proteins were abundant and hypertrophic markers were at low levels [128].

In most cases, strategies to demonstrate the ability of ADSCs to undergo chondrogenic differentiation incorporate a high-cell-density culture supplemented with TGFβ1. Hypoxic cell expansion is a widespread practice to facilitate chondrogenic differentiation too. ADSC pellets differentiated in a chondrogenic medium with dexamethasone and TGFβ1 after hypoxic expansion (1% of O_2_) increased the production of the chondrogenic markers (SOX9, COL2) in vitro, but not to the extent that BM-MSC pellets did [129]. At the same time, the hypoxic expanded ADSC- and BM-MSC-derived chondrogenic pellets showed *COL10* gene up-regulation, both in vitro and, after intra-articular injection, in an animal knee defect model [129].

The culturing of ADSCs within slowly degraded gelatin scaffolds, organized like hyaline cartilage lacunae [130,131], led to up-regulation of SOX9 and COL2 expression and GAG production two weeks after chondrogenic induction compared to randomly organized scaffolds. As a result, it was suggested that the scaffold geometry influenced the chondrogenic differentiation of ADSCs [130]. Additionally, MRI inspection, histological examinations and IHC staining of the ADSC-seeded organized scaffold samples implanted subcutaneously into nude mice showed a high chondrogenecity potential in implants [130].

### 6.3. Alternative Strategies for Treating Joint Defects with ADSCs

An alternative strategy for treating joint defects involved impeding the terminal hypertrophic differentiation of ADSCs using the ECM protein matrilin-3 (MATN3) [132] as a component binding to BMP2 and preventing BMP2 downstream signaling, required for *Col10* expression in chondrocytes [133]. ADSCs encapsulated with Matn3 in hydrogel showed favorable effects by increasing the chondrogenic marker and decreasing the hypertrophy marker mRNA and protein expression in vitro. In vivo, MATN3-loaded ADSC spheroids prevented subchondral bone sclerosis in a chronic OA mouse model [132].

Recently, a strategy was proposed to simplify the cell therapy procedure and remove steps associated with the enzymatic isolation, cell expansion and scaffold fabrication. Autologous fractionated adipose tissue (so-called ECM/SVF-gel in the study), a gel-like substance consisting of adipose tissues mechanically destroyed by simple mechanical shifting between two syringes, was applied to cartilage injury repair in combination with a microfracture in a rabbit model [134]. Twelve weeks after the ECM/SVF-gel injection, the knee joints were evaluated using MRI, macroscopic observation, histology and IHC. It was reported that the ECM/SVF-gel could improve articular cartilage regeneration, integration with surrounding normal cartilage and the expression of COL2 compared to the microfracture treatment alone [134].

Despite the achieved progress of ADSC applications in clinical studies, cell therapies involving ADSCs require further method modifications to improve differentiation efficacy.

## 7. Periosteum- and Perichondrium-Derived Stem Cells

Skeletal long bone formation starts from chondrogenic cell condensation and is subsequently transformed into differentiated chondrocytes surrounded by a thin tissue sheath, termed the perichondrium [135]. The terminal step of the chondrocyte differentiation process is hypertrophy, after which the cartilage is replaced by bone tissue. The periosteum is the new sheath of flattened, elongated cells surrounding the bone region. The perichondrium and periosteum are reported to have similar morphology, composed of two layers, the outer fibroblastic layer and the inner cambial layer, containing a niche of pluripotent stem cells with both chondrogenic and osteogenic capabilities of differentiation [136,137]. The collagen fiber orientation in the perichondrium and periosteum concurs with the assessed directions of tissue growth [138]. In situ hybridization analysis of genes from embryonic-day-12 tissue revealed that the perichondrium and periosteum are distinct tissues at the molecular level [139]. Both the perichondrium and periosteum participate in limb growth regulation by serving as sources of signaling molecules involved in proliferation, differentiation and hypertrophy of the chondrocytes and respond to signals from underlying cells [140].

### 7.1. Periosteum

#### 7.1.1. Periosteal Adult Stem Cell Populations

The periosteum is a double-layer thin membrane that is well vascularized and innervated. It envelopes nearly every bone in the body [141], except for the intra-articular surfaces and sesamoid bones [142]. The inner cambium layer of the periosteum contains a heterogeneous cell population, including fibroblasts, endothelial cells, mast cells, pericytes and several specific periosteal stem cell (PSC) populations that have been recently identified [143,144,145,146], each with distinct physiologic functions [144]. PSCs display clonal multipotency and self-renewal [144,147]. Single-cell and bulk transcriptional profiling shows that PSCs are distinct from other skeletal stem cells and mature mesenchymal cells [144]. The periosteum plays the role of being a reservoir of osteoblasts and osteochondroprogenitors [144,148,149]. It is responsible for the radial bone growth and turnover of the bone cortex throughout life and orchestrates the healing of bone fractures [143,144,146]. Notably, the osteoblastic potential of the periosteum differs with location [150], with the calvarial periosteum demonstrating less osteogenic potential than the periosteum covering the tibia [151].

Modern genetic engineering techniques allow for cell labeling in the periosteum for detailed research of periosteal adult stem cell populations in vivo and in vitro. The lineage tracing method demonstrated that the adult periosteum contains αSMA-positive cells that are long-term, slow-cycling, self-renewing osteochondroprogenitors, identified by cell sorting and scRNAseq methods as different mesenchymal populations to those present within the bone marrow [149]. Using a fluorescent reporter adult animal model, most Prx1-labeled cells of the periosteal cambium layer were reported to be periosteal osteochondroprogenitor cell populations converting to osteogenic cells in response to mechanical stimulation, directly sensed via the primary cilia [152]. Changes in the mechanical properties, such as loss of the periosteum’s baseline stress, result in significantly increased crimping of collagen in the periosteum’s fibrous layer and changes in the nucleus shape of cells in the cambium layer. These signals influence the equilibrium of the periosteal stem cell niche and change the fate of the periosteum-derived cells [153].

#### 7.1.2. Periosteal Graft Transplantation

The efforts of periosteal graft transplantation began a relatively long time ago. In 1982, a rabbit study investigated articular cartilage defects using periosteal grafts and, after one year, the defects were filled with white glistening tissue, very similar to normal hyaline cartilage [154]. Research on patients followed clinically through arthroscopic examination and radiography showed similar outcomes [155]. Unfortunately, the long-term outcomes (6–9 years) of periosteal transplantation deteriorated with time and the method failed to produce hyaline cartilage and prevent the development of arthrosis [156]. The promising results after periosteal grafting obtained in 1980 could be due to the young age of the rabbits (6 months old) [154] and patients (16–34 years old) [155] and the lack of techniques for estimating results that are available now. As a result, studies on rabbits by O’Driscoll et al. [157] indicated better regeneration when the periosteal graft was sutured into the defect with the cambium layer facing the joint space, not the subchondral bone. Under continuous passive motion after transplantation, the periosteal grafts produced tissue containing predominantly Col2 [157].

For the ACI method, during arthrotomy, the joint defect was covered with a periosteal flap taken from the proximal medial tibia, under which chondrocytes were injected [158]; however, the results of transplantation with the periosteum were inconsistent. Additionally, the periosteal tissue was too thin and fragile for surgical manipulations [159]. In the clinic trials, the incidences of transplant hypertrophy were reported to be the highest in the group of patients treated with periosteum-covered ACI compared to other ACI technique variations [160].

Recently, an adult periosteum was reported that had a SOX9-positive skeletal progenitor population capable of giving rise to osteoblasts, mature cortical osteocytes and chondrocytes [161]. However, applying periosteum grafts for cartilage repair retains a risk of subsequent ossification [162] through endochondral mechanisms, as it naturally happens during long bone fracture healing [147]. To understand the tissue perfusion impact on the periosteum graft ossification along with cartilage regeneration, a vascularized pedicled periosteal flap and a non-vascularized periosteal graft were applied to a rabbit knee defect model for eight weeks, with anti-ossification reagents, such as fulvestrant and IL1b, injected every two weeks to prevent bone formation [163]. Improved defect regeneration was observed in the vascularized periosteum flap group with a higher COL2 expression and proteoglycan content compared to the non-vascularized periosteum group and the groups with anti-ossification medication [163].

Theoretically, periosteal flap transplantation could be adapted so it can be performed arthroscopically. However, the periosteum alone has been shown not to repair defective hyaline cartilage. Adult human periosteum-derived cell cultures have a high proliferation rate with stable retention of the TGFβ1-induced chondrogenic differentiation potential along 15 passages independently of donor age [164]. Such in vitro characteristics have made stem cells from the periosteum an interesting precursor cell source for tissue engineering constructs. Key aspects of developing methods for repair of articular cartilage could potentially be based on tissue engineering structures seeded with chondrocyte precursors from the cambium layer.

### 7.2. Perichondrium

The perichondrium is a thin layer of dense connective tissue surrounding the nasal, costal and tracheal hyaline cartilage and the auricular elastic cartilage in adults. It leaves only the articular cartilage surfaces uncovered. The first investigations of the chondrogenic regeneration potential of the adult perichondrium occurred during the 1970s [165,166,167,168]. At that time, costal perichondrium transplantation reconstruction had been used for articular surfaces in small-finger joints damaged by infection or trauma. Decades later, an observational study evaluating the long-term results was published [169]. The injured finger joints restored by perichondrium transplants remained essentially pain free and mostly well functioning without the need for additional surgeries up to 41 years after the procedure [169], which encourages us today.

#### 7.2.1. Perichondrium Transplantation

About twenty years ago, chondrogenic stimulation of perichondrium grafts in vitro promoted by synovial fluid collected from non-inflammatory joints was shown. Then, costal and auricular perichondria from adult rabbits were transplanted intra-articularly and intra-abdominally for eight weeks [170]. The morphology analysis of graft sections demonstrated that better chondrogenesis-like processes were found in rib perichondrium specimens placed in the knee joint, in contrast to grafts placed intraperitoneally. However, the scarcity of analysis methods in the 1990s meant that representative data for this outcome could not be provided [170].

The costal perichondrium contains a unique niche housing pro-chondrogenic cells, contributing to costal cartilage regeneration [171]. The rib perichondrium and periosteum were extracted from a rat with green fluorescent protein (GFP) transgene expression and transplanted into articular cartilage defects of regular laboratory rats [172]. Tracing of the GFP cells showed newly formed tissues that originated from the perichondrial or periosteal transplants, which acquired chondrocyte marker expression patterns at day 112 post-surgery. However, in contrast to the perichondrium, the transplanted periosteum expressed COL1 throughout the experiment and lost SOX9-expressing cells two weeks after the surgery [172].

In 2002, clinical applications of perichondrial grafts to isolated cartilage defects did not show advantages compared to the drilling procedure in the 10 years of follow-up. Irregularities, such as calcifications of the graft, subchondral sclerosis or pain, were seen to the same extent in both groups [173]. New reports regarding transplantation of perichondrium flaps/grafts into articular cartilage defects have not yet emerged.

#### 7.2.2. Perichondrium-Derived Adult Stem Cells

The perichondrium has been used as a source of stem cells for follow-up cell therapy procedures. In vitro tests of auricular PchDCs obtained through fibronectin adhesion showed cell expression of MSC-appropriate positive surface markers, high colony-forming efficacy and proliferative ability and mesenchymal trilineage differentiation potential [174]. PchDCs isolated by collagenase digestion from auricular and tracheal perichondria demonstrated high proliferation and hyaline cartilage-like matrix protein production according to histology and IHC assessments [175]. Chondrogenic stimulation resulted in porcine pellets of auricular or tracheal PchDCs returning high expression rates of COL2 and ACAN without a trace of COL10 and elastic fibers [175].

Cells grown from explants of rabbit rib perichondria in culture media with TGFβ1 were shown to enhance proliferation rates and *Col2* gene expression [176]. Similar results were demonstrated for cultures of aged perichondrium-derived cells (PchDCs) [177].

Finally, another research strategy to consider is the transplantation of rat rib PchDCs infected by adenoviral vectors containing sequences of BMP2 and/or IGF1 [178]. Cells producing both growth factors placed into a joint defect generated repair tissue with a hyaline morphology and high proteoglycan and COL2 contents. Transplantation of uninfected perichondral cells in the control group resulted in insufficient repair two weeks after the surgery. However, transplantation of BMP2-producing cells could lead to osteophyte formation if they partially dislocate to the joint margins [178].

Considering the above data, research undertaken with high caution and responsibility may develop cell therapies for restoring hyaline cartilage based on periosteal cells. Currently, the perichondrium has lost the interest of researchers as a cell source for regeneration therapies.

## 8. Dental Pulp Stem Cells

In the central cavity of each tooth, there is an unmineralized tissue, dental pulp, which contains vast numbers of different cell types, including a stem cell niche [179]. Dental pulp stem cells (DPSCs) originate from neural crest cells and demonstrate MSC features, such as colony formation in vitro [180], a rapid proliferation rate, a strong capacity to self-renew and multiple differentiation potentials [181,182]. DPSCs have the classical MSC surface markers and lack CD11b, CD14, CD19, CD34, CD45, CD79a and HLA-DR transmembrane proteins [182,183]. The DPSC harvesting method is minimally invasive [184,185,186]. DPSCs are obtained from collected teeth by enzymatic digestion or by the explant method [187]. Additionally, DPSCs have impressive cryopreservation abilities and are suitable for banking stem cells collected after routine teeth extraction [188].

### 8.1. Conversion of DPSCs to Chondrocytes

DPSCs have been applied in articular hyaline cartilage tissue regeneration and tissue engineering [189]. It was demonstrated that human DPSCs suppressed OA macrophage activation in a direct co-culture system in vitro and attenuated damage to the articular cartilage in a rabbit knee OA model in vivo [190]. Local injection of DPSCs in a model of rats with temporomandibular joint (TMJ) arthritis relieved hyperalgesia and synovial inflammation and attenuated cartilage matrix degradation, according to histologic staining and μCT scanning results [191].

Attempts to directly stimulate the conversion of DPSCs into chondrocytes have also been made. Chondrogenic differentiation of DPSCs in vitro was achieved using a differentiation medium with ITS and ascorbic acid. DPSCs acquired a more rounded morphology and had higher amounts of COL2 and ACAN, with no expression of COL1. However, it could not reach the same expression levels of ECM proteins obtained by primary articular chondrocyte cultures in the same differentiation medium [192].

In a recent in vitro study, Longoni et al. cultured DPSC pellets for 21 days in a chondrogenic medium with ITS and dexamethasone supplemented with different growth factors, such as TGFβ1, TGFβ3, BMP2, BMP6, BMP7 and IGF1, in various combinations. Nevertheless, cells in the pellets had a spindle fibroblast-like morphology and did not form typical hyaline cartilage lacunae and the ECM mostly had COL1 without COL2 and ACAN expression [181]. To improve the chondrogenic differentiation outcomes, human DPSCs were cultured in a 3D organization based on 3% agarose wells in a commercial chondrogenic medium supplemented with TI, TGFβ1 and a recombinant analog of IGF1. The synthesis of ACAN, COL2 and less COL1 was then immunodetected in microtissues formed by DPSCs [193].

Some peculiar efforts have been undertaken to improve the chondrogenic differentiation of DPSC cultures. The chondrogenesis-related genes were significantly up-regulated and GAG synthesis was increased in the DPSC monolayer on day 21 when the chondrogenic medium was accompanied by taurine. Elastin expression in the culture was also observed [194]. To improve the chondrogenic differentiation of the human DPSCs, an extract of sonicated lactic acid bacteria was added to the chondrogenic medium, including ITS, dexamethasone and TGFβ1 [195]. This extract enhanced the expression of early stage chondrogenic marker genes and hypotrophic marker genes for the first week in DPSC pellet cultures compared to the control group [195].

### 8.2. TGFβ3 as Chondrogenic Initiator for DPSC Cultures

More impressive results have arisen from research regarding the co-culturing of human costal chondrocytes (CCs) with human DPSCs in pellets [196]. Data demonstrated that CCs could supply a chondro-inductive environment that promoted the chondrogenic differentiation of DPSCs. However, CCs alone could not stop the mineralization of the pellets, so the exogenous growth factor FGF9 was added to prevent hypertrophic differentiation and to stabilize the permanent cartilage-like phenotype of the DPSC and CC co-culture. The co-culture pellets with added FGF9 in a chondrogenic medium with TGFβ3 demonstrated more intense Alcian blue and COL2 IHC staining. In contrast, the DPSCs cultured in regular DMEM formed only fibrous tissues and the DPSCs cultured in the chondrogenic medium with TGFβ3 but without FGF9 showed the highest expression level of COL10 [196]. The same study showed that biodegradable PGA scaffolds seeded with DPSCs preliminarily co-cultured with CCs formed homogeneous cartilage-like tissue with a similar structure and high matrix deposition eight weeks after transplantation into the backs of nude mice [196].

Transduced human DPSCs with the highly expressed chondrogenic growth factor TGFβ3 were seeded on a poly-l-lactic acid/polyethylene glycol electrospun fiber scaffold and implanted under the skin in nude mice [197]. Under TGFβ3 influence, transplanted DPSCs showed the presence of chondrogenic markers (COL2, SOX9 and ACAN), as verified by IHC staining and mRNA level measurement. An in vivo study of human DPSCs seeded onto a COL2/COL3 scaffold in a mini-pig knee defect model was conducted, but there were no representative results [198].

DPSCs have potential as an alternative source of adult MSCs for joint cartilage regeneration. However, there are still no valuable pre-clinical or clinical trial results using DPSCs.

## 9. Extracellular Vesicles (EVs) as a Tool for Cell Fate Manipulation

Another way to manipulate the fate of cell populations is to use extracellular vesicles (EVs), first discovered in the last century [199,200]. EVs are a heterogeneous population of lipid bilayer-enclosed particles produced by different cell types as an essential carrier of cell-to-cell communication mediators [201], implemented through the transfer of a variety of proteins, lipids and RNA cargos. Due to their unique biological roles, EVs coordinate complex processes, such as development, inflammation, tumorigenesis and even maintenance of a stem cell niche [201]. EVs release responses to external influences, such as cyclic hydrostatic pressure [202], oxidative stress [203] and hypoxia [204]. Types of EVs are distinguished for subclasses by their fundamental mechanisms of biogenesis, diversity of their composition and size [205] and selective cargo loading [206,207]. The major subclasses of EVs are composed of membrane-derived vesicles called microvesicles (MVs) (100–1000 nm) as a result of the outward budding of the plasma membrane and exosomes (50–150 nm) originating from endosomal multivesicular bodies [205,208].

Most commonly for research purposes, EVs are isolated from a culture medium using centrifugation and filtration methods, enzyme-linked immunosorbent assay (ELISA) and different exosome isolation kits that are commonly found. However, the homogeneity of the isolated particles remains low and current methods cannot efficiently separate the different types of EVs extracted [209]. Over the past two decades, EVs have attracted intense interest regarding their capability to regulate stem cell self-renewal and differentiation [201]. It was demonstrated that soluble mediators released by stem cells, rather than the cells themselves, could provide significant therapeutic benefits [208,210] that open up great opportunities for their application in regenerative medicine.

### 9.1. Chondroprotective Effects Mediated by EVs

The immunomodulatory capacity of EVs has been extensively recognized. Using in vitro models of OA based on chondrocytes treated with pro-inflammatory factors, the immunosuppressive properties of EVs derived from BM-MSCs [211,212], ADSCs [213] and embryonic stem cell cultures [214] were shown to decrease the expression of catabolic (ADAMTS5 and MMP13) and pro-inflammatory markers (COX2, iNOS and interleukins) and increase levels of COL2 and ACAN. These outcomes were probably mediated by the inhibition of the nuclear factor kappa-light-chain-enhancer of activated B (NF-kB) pathway [215].

EVs derived from TGFβ3-pretreated BM-MSCs exerted chondroprotective and anti-inflammatory functions in OA-like murine chondrocytes, confirmed by re-inducing the expression of chondrocyte markers while inhibiting catabolic (MMP13, ADAMATS5) and inflammatory (iNos) markers in vitro [210]. Additionally, intra-articular injection of BM-MSC-derived EVs had a moderate protective effect on cartilage and bone from OA damage in a collagen-induced arthritis model in mice [210].

Studies involving DPSC-derived EVs are much less common. A single intravenous injection of conditioned medium (CM) from human DPSCs in a mouse model of RA markedly attenuated tissue destruction of the ankle joint compared to the control and even the group with intravenous injection of CM from BM-MSCs [216].

### 9.2. EV-Mediated Chondrogenesis

EVs can promote chondrogenesis through cell-to-cell transfers of EVs’ natural cargo. Data obtained from a rabbit TMJ OA model also confirmed that human BM-MSC-derived EVs could promote cartilage reconstruction mediated by increasing factors related to cell proliferation (PCNA protein) and cartilage matrix formation (COL2, ACAN) [217]. Recently published proteomic analysis of EVs obtained from adipose tissue, BM and synovium revealed their distinct protein profiles [218]. Simultaneously, BM-MSC culture studies reported comparable chondrogenic effects implemented by EVs of different origins. According to gene ontology (GO) analysis, the three types of EV proteomes differentially regulate biological processes, such as ECM organization, cartilage condensation, cell growth and cell migration. Additionally, there was no apparent difference among the three EV-treated groups of BM-MSCs encapsulated in Matrigel with the chondrogenic factor TGFβ3 in a nude mouse subcutaneous transplantation model [218].

### 9.3. Exosomal microRNAs as Therapeutics

Mechanisms of EVs’ actions of various origins are not known for certain. However, an important role is increasingly evident in the presence of various regulatory microRNAs, which affect different signaling pathways of target cells. To date, it has become possible to use genetically engineered or chemically modified exosomes, a subclass of EVs, for targeted delivery of therapeutics to specific types of cells or tissues [219]. For instance, exosomes derived from chondrocyte cultures induced chondrogenic differentiation of BM-MSCs by transferring exosomal miR-8485 to regulate the Wnt/β-catenin pathway [220]. Equally, exosomes isolated from rabbit chondrocytes demonstrated stimulation of chondrogenesis with minimal hypertrophy in ACPCs harvested via differential adhesion to fibronectin and implanted subcutaneously into mice [221]. In vitro treatment of hBM-MSCs with exosomes derived from miR-95-5p-overexpressing human chondrocytes promoted cartilage-specific gene expression in TGFβ3-induced chondrogenesis [222].

The response of human chondrocyte cultures to SM-MSC-Exo application was effective proliferation and migration. However, the drawback was the striking inhibition of *Sox9* and its downstream genes, including *Acan* and *Col2*, via the WNT5a and WNT5b factors enriched in exosomes [223]. This side effect was blocked through exosomes isolated from SM-MSCs transfected with miR-140-5p identified as a cartilage matrix modulator. SM-MSC-Exos with miR-140-5p enhanced the proliferation and migration of chondrocytes without damaging SOX9, ACAN and COL2 secretion in vitro. They mitigated the severity of the damage to knee articular cartilage in a rat OA model in vivo [223].

The application of EVs yields a promising, cell-free field of regenerative medicine without apparent adverse effects, such as immunogenicity or tumorigenicity. However, in addition to the unclear molecular mechanism of EVs outcomes, the formal difficulties associated with EV heterogeneity, lack of unified isolation protocol and method of quality attestation need to be addressed.

## 10. Conclusions

Today, the idea of applying autologous adult stem cells for cartilage regeneration is the most-developed concept in regenerative medicine. The different types of adult stem cells obtained from variable tissue sources have different advantages in accessibility, immunogenicity, proliferative and chondrogenic abilities. Clinical trials of intra-articular injections of undifferentiated autologous stem cells did not provide the desired outcomes; in most cases, there was no significant difference in efficiency between injected stem cells and surgical practices, such as microfracturing.

Different stem cell populations respond to chondrogenic stimuli in various ways, so strategies initiating chondrogenicity could vary from one adult stem cell population to another. Very likely, there is no unique differentiation method suitable for all or several stem cell populations of a patient. Chondrogenic factors, such as growth factors, regulatory molecules, the geometry of the scaffolds used, cell density seeded and physical stimuli, can independently control the fate of stem cells but can give a synergistic effect in combinations. Heterogeneity within the adult stem cell population from one tissue source makes more complicated the assessment of the chondrogenic efficiency of these factors. It is necessary to determine the specific markers that separate the internal cell subsets of one type of stem cell according to their chondrogenicity and obtain the optimal ones. Single-cell transcriptome sequencing has become a technique helping to perform cell clustering based on specific markers and compare cell subsets defined. The development of various external options to guide the fate of stem cells enables the evolution of joint damage therapies and investigates the mechanisms of cellular interactions.

## Figures and Tables

**Figure 1 ijms-23-11169-f001:**
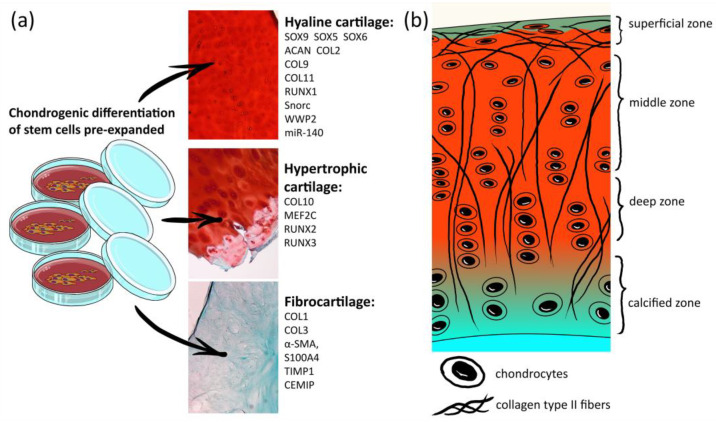
List of markers for hyaline cartilage chondrocytes, hypertrophic chondrocytes and fibrocartilage. (**a**) Histological images are photos of human knee articular cartilage sections stained by Safranin-O/Fast green; (**b**) scheme of the human articular cartilage structure.

**Figure 2 ijms-23-11169-f002:**
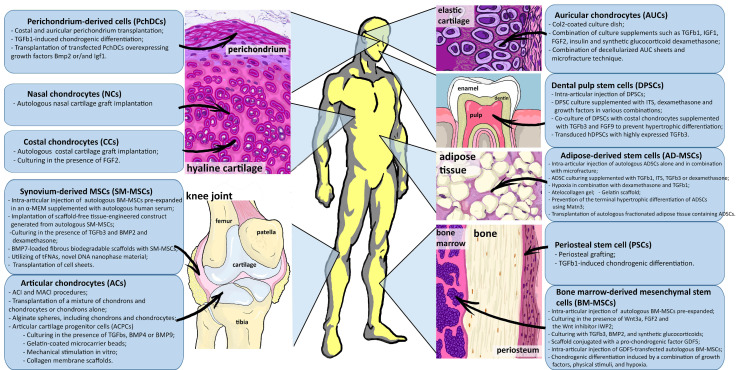
Schematic summary of the current review discussing distinct approaches of autologous adult stem cell chondrogenic differentiation to repair hyaline cartilage damage.

## Data Availability

Not applicable.

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
