# Peer review of "Strategies to Convert Cells into Hyaline Cartilage: Magic Spells for Adult Stem Cells"

_ijms, 2022, doi:10.3390/ijms231911169_

Round 1

Reviewer 1 Report

This work by Kurenkova and col. deals with a very interesting issue in the field of regenerative  medicine, which is efficient cartilage regeneration. The manuscript is overall well written and easy to follow, and the subject is interesting for the scientific community. However, I believe there are some modifications that, if introduced by the authors, will increase the quality of their manuscript.

This work is quite comprehensive and deals with many different aspects of cartilage regeneration, but I have found it to be quite long. I believe it can be shortened by avoiding writting about well stablished facts in the field. For example, all experts in this field know the ACI and MSCI techniques so, there is no need for ample explanations. The same goes for the definition of mesenchymal stem cells or the pathways involved in their differentiation. Also, some text could be substituted by a figure, for example, the description of the inner structure of the cartilage. This would lighten up the manuscript. The manuscript should be reduced as much as possible.

Some concepts such that of the chondron, the relation between synoviocytes and SM-MSCs (this is indeed quite confusing) or that of ACPCs should be clarified. In particular, the authors write indistinctly about ACPCs, progenitor cells and MSCs. This is confusing. The ontogeny/origin of these types of cells as well as the relation between them should be clearly explained. A figure might help with this.

One of the main flaws of this manuscript is that it lacks structure. The authors mix in the same section treatments based on the local injection of cells and the use of those same cells in conjunction with scaffolds. There is no need to make multiple subsections but a couple of those would certainly help to organise the manuscript. For instance, in the section where the authors talk about the different types of cells used in cartilage regeneration, there should be a subsection describing the cells and another subsection/s describing some of the experimental settings where they have been used and the outcome. Also, if different approaches are explained (i.e. intra-articular injections or scaffold based techniques) they should be structure in further sections. Again, this part can be importantly reduced.

Some concepts are a bit outdated, for example, the markers that define MSCs are constantly updated. The authors should take into consideration the latest communications of the international society for stem cell in this regard. In general , I find the references are generally not very up-to-date. Since this field is in constant evolution, an effort should be done to use references from the last 5 to 10 years. Seminal works are obviously excluded of this comment.

The authors refer to "pellets" at different points in the manuscript but I have not seen that they explain what those pellets are. This should be briefly explained for those that are not experts in the field.

In their last section the authors refer indistinctly to exosomes and extracellular vesicles. That is fundamentally wrong. The authors should clearly explain that exosomes are a subclass of extracellular vesicles, and that the latest findings about the role of miRNAs delivered through EVs refer almost exclusively to exosomes delivered miRNAs. Please, clarify this.

Reviewer 2 Report

IJMS-1906568

Strategies to convert cells into hyaline cartilage: magic spells for adult stem cells

This review article summarizes recent stem cell-based strategies in the treatment of osteoarthritis (OA) to restore damaged hyaline cartilage. The composition of different forms/layers of cartilage, markers for the respective chondrocytes, and the challenge of adequately repairing damaged cartilage are clearly described. Sources, isolation methods, and handling/differentiation of autologous chondrocytes as well as stem cells derived from different tissues are well presented and illustrated. Classical as well as the latest in vitro and in vivo approaches including their pros and cons are discussed.

Text and figures appear straightforward and clear. The article adequately covers the topic, reflects the relevant parts of the literature (including the latest reports), and provides a broad and informative overview for the reader. Therefore, only a few aspects/questions have to be addressed.

1.      The manuscript (including the references) has to be formatted according to the IJMS guidelines.

2.      Greek letters have to be used where necessary.

3.      Page 3: Please describe in short how “… the hypertrophic and fibrocartilage chondrocyte phenotypes are associated with OA development”.

4.      Page 4: Please provide more details on the role of Snorc in chondrocyte biology.

5.      A few comments on the relationship of fibrosis and fibrocartilage in OA should be included.

6.      Are negative isolation protocols available for the isolation of the different (stem) cells used?

7.      Pages 10, 14: Is there a general shift in gene expression induced by hypoxia that may be detrimental to the cells (or their further use) which have been expanded under these conditions?

8.      Page 11: Which “… specific markers to separate the synovial membrane-derived MSCs … according to their location in the synovium …” have been identified?

9.      Page 15, “It was reported that ECM/SVF-gel could facilitate articular cartilage regeneration compared to the microfracture treatment alone”: Did the ADSCs in this approach express markers of chondrogenic differentiation?

10.   Page 16, “Changes in the mechanical properties, …, influence the equilibrium of the periosteal stem cell niche and change the fate of the PDCs”: Please better describe the relationship between mechanical stress and the equilibrium of the stem cell niche or PDC fate, respectively.

11.   Page 19, “… cells in the pellets had a spindle fibroblast-like morphology …”: Did these cells also express (surface) markers of fibroblast-like cells?

12.   Page 19, “… transplanted DPSCs showed the presence of chondrogenic markers, …”: Please provide the respective markers.

13.   Page 20: (i) “It was demonstrated that factors released by stem cells, …”, (ii) “… decrease pro-inflammatory marker expression and increase chondrogenic marker levels”, and (iii) “… increasing factors related to … cartilage matrix formation”: Please provide a few examples for the factors/markers mentioned.

14.   Page 20, “… analysis of exosomes … revealed their distinct protein profiles…”: Please comment on the factors supporting chondrogenic development included in the exosomes.

Round 2

Reviewer 1 Report

This reviewer has no further comments.